# Association of gestational weight gain rate with infant birth weight and cesarean delivery: A prospective cohort study in Nepal

Jyoti Nepal[1], Kalpana Chaudhary[1,2], Bikram Adhikari[3], Abha Shrestha[4], Archana Shrestha[1,2,5], Sangita Pudasainee Kapri[6], Shristi Rawal[7] *

1 Research and Development Department, Dhulikhel Hospital-Kathmandu University Hospital, Dhulikhel, Bagmati, Nepal, 2 Department of Research, Institute for Implementation Science and Health, Kathmandu, Bagmati, Nepal, 3 Department of Research, HERD International, Kathmandu, Bagmati, Nepal, 4 Department of Obstetrics and Gynecology, Dhulikhel Hospital-Kathmandu University Hospital, Dhulikhel, Bagmati, Nepal, 5 Department of Chronic Disease Epidemiology, Yale School of Public Health, New Haven, Connecticut, United States of America, 6 School of Nursing, Rutgers University, Camden, New Jersey, United States of America, 7 Department of Clinical and Preventive Nutrition Sciences, School of Health Professions, Rutgers University, Newark, New Jersey, United States of America

* shristi.rawal@rutgers.edu

**Data Availability Statement:** The data underlying this study cannot be made publicly available due to

## Abstract

Gestational weight gain (GWG) is linked to pregnancy outcomes, such as birth weight and delivery mode, though research in low-income countries like Nepal is limited. We examined the association of GWG rate with infant birth weight and cesarean delivery in a prospective cohort of 191 pregnant women in Nepal, using data collected from August 2018 to August 2019 at a peri-urban hospital in Dhulikhel. Participants included women with singleton, full-term live births, with GWG rate calculated from weight gain between the second and late pregnancy stages, divided by the weeks in between. GWG rate categories—adequate, inadequate, or excessive—were defined by pre-pregnancy Body Mass Index (BMI) specific to GWG recommendations from the 2009 Institute of Medicine report. Ethical approval was obtained from Institutional Review Board of Kathmandu University and Rutgers University. Statistical analyses in SPSS and Stata revealed that 52.4% of mothers exceeded the recommended GWG rate, particularly among overweight and obese women (0.4 ± 0.2 kg/week and 0.5 ± 0.2 kg/week, respectively). The average birth weight was 2964.9 ± 407.0 grams, with 12% of infants classified as low-birth-weight. Cesarean delivery was recorded in 45% of the women. After controlling for factors like age, education, ethnicity, occupation, parity and BMI, each 1 kg/week increase in GWG from the second to third trimester correlated with a 392-gram increase in birth weight (β = 391.9, 95%CI = 67.2–716.7, p = 0.01), while excessive GWG rate led to a 148-gram increase over adequate GWG rate (β = 148.1, 95%CI = 8.7–287.5, p = 0.03). However, GWG rate was not significantly linked to cesarean delivery. These findings suggest that maternal GWG rate positively affects infant birth weight but not cesarean delivery, underscoring the need for larger studies to explore GWG rate's effects on maternal and neonatal outcomes.

compliance with the Institutional Review Board (IRB) of Rutgers University-approved protocol and consent form, which does not have a provision for sharing the data publicly with external investigators. The Rutgers University IRB can be contacted about further information on data requests at eirb@research.rutgers.edu.

**Funding:** This work was supported by the Rutgers Global Health Institute and the National Institutes of Health/FIC (grant number 1R21TW011377-01 to SR) awarded to Shristi Rawal and Archana Shrestha. The funders had no role in study design, data collection and analysis, decision to publish, or preparation of the manuscript.

**Competing interests:** The authors have declared that no competing interests exist.

## Introduction

Globally, the impact of gestational weight gain (GWG) has gathered significant attention as various studies showed that pregnancy outcomes are adversely affected by inadequate or excessive GWG which jeopardizes the health of both the mother and the unborn child [1–3]. In 2009, the US Institute of Medicine (IOM) studied pregnancy outcomes in relation to prenatal body mass index (BMI) and GWG and established guidelines for GWG goals [4].

Numerous studies done in Asian women that looked at the correlation between GWG and short-term maternal and neonatal outcomes, found that women with excessive GWG had a higher risk of macrosomia, large for gestational age, and cesarean section compared to those gaining weight within the suggested guidelines, whereas those with inadequate GWG had a higher risk of delivering low birth weight, small for gestational age, and preterm born infants [5–9].

Birth weight is a key indicator of a baby's health and there are very few studies in Nepal that have assessed the maternal factors that determine the birth weight of neonates including maternal weight gain during pregnancy [10]. Nepal has had a rising rate of cesarean delivery over the past decades [11]. The percentage of live births delivered by cesarean-section (CS) has shown a steady increase over the years, rising from 1% in 1996 to 10% in 2016 and reaching 18% by 2022 [12], Understanding the role of GWG in this context is necessary where limited studies have been conducted [13,14]. With the double burden of communicable and non-communicable diseases in Nepal, examining the pattern of GWG and its relationship between maternal and newborn outcomes is crucial. We, therefore, aimed to prospectively examine the association of GWG rate (using IOM weight gain guidelines) with infant birth weight and cesarean delivery among a Nepalese pregnancy cohort.

## Materials and methods

### Study design, duration and setting

This study is a hospital-based prospective cohort study. The recruitment of participants started from 15th August 2018 and the data collection continued till 20th August 2019 at Dhulikhel Hospital-Kathmandu University Hospital in Nepal. Located 20 kilometers outside of the capital city, Kathmandu, Dhulikhel Hospital is a community-based tertiary level hospital which has a catchment population of 1.9 million people and delivers approximately 3,500 babies annually.

### Study participants

A total of 244 pregnant women participated in the study. Women receiving antenatal care from the Obstetric Outpatient Department at Dhulikhel Hospital, women with ≤14 weeks gestation at the time of enrollment, singleton pregnancy and age 18 years and older were eligible for the study. Overall, 191 participants who had all the required antenatal visits with liveborn, term delivery were included in the analysis. Taking reference to a retrospective study conducted in China [15], our sample size had more than 90% power to assess GWG rate and its association with neonatal birth weight.

### Ethical approval

The ethical approval for the study was obtained from the Institutional Review Board of Kathmandu University (102/18) and Rutgers University (Pro2018001976). We obtained written informed consent from each participant before data collection. The confidentiality of each participant was maintained and voluntary participation was ensured. Participants under the age

of 18 years were not included in the study. The study protocol and conduct adhered to the principles in the Declaration of Helsinki.

## Study measures

Face to face interviews were conducted by two trained research assistants of Dhulikhel Hospital to obtain the socio-demographic information (age, ethnicity, religion, education, occupation and income), gravidity and parity. Trained research nurses collected data on anthropometric variables such as height, body weight at first, total weight gain in pregnancy as well as second and third trimester visit. Medical records were used to obtain information on neonatal outcomes (birth weight, birth length, head circumference, chest circumference and gender) and maternal outcomes (mode of delivery, gestational weeks at delivery and pregnancy complications). GWG rate was reported as the rate of weight gain from the second trimester to late pregnancy and was calculated by subtracting the measured weight at the second trimester (13–25 weeks gestation) from the measured weight at the third trimester (28–35 weeks gestation), and dividing this by the number of weeks in between. The first trimester weight was used as a proxy for pre-pregnancy BMI. The adequacy of GWG rate was categorized as inadequate, adequate, or excessive based on gestational weight recommendations from the 2009 Institute of Medicine report [4].

## Statistical analysis

Data was collected in Kobo Toolbox exported to IBM SPSS version 20 where checking, cleaning, editing and analysis of the data was performed. Descriptive analysis was done to report frequency, percentage, mean and standard deviation. The income was categorized as above or below the median. Linear regression was performed to determine the association of neonatal birth weight with GWG rate and binary logistic regression was performed to determine the association of cesarean delivery with GWG rate. Continuous propensity scores derived from Stata 14, were used to address potential confounders such as age, education, ethnicity, occupation, parity, and pre-pregnancy BMI. Directed Acyclic graph (DAG) of potential confounders in relation to the study variables is shown in Fig 1.

## Results

### Characteristics of the study population

Table 1 presents the characteristics of the study participants. Participants' mean age was 26.0 ± 4.2 years. Brahmin/Chettri and Newar were the largest ethnic groups (40% and 37%, respectively). The majority of participants (88.5%) identified as Hindu. Average schooling was 12.4 ± 3.4 years. Half were homemakers, and 52% reported a monthly income more than $229. Regarding pregnancy history, 43% were primigravida and 56% were nulliparous. Gestational diabetes affected 4%, and 11.5% had gestational hypertension. The mean gestational weeks at delivery was 38.5 ± 1.1 weeks. The participants' average pre-pregnancy BMI was 24.6 ± 3.7 kg/m$^2$. The majority of participants (60.2%) had normal BMI status prior to pregnancy.

### Gestational weight gain rate

In our group of participants, the average rate of additional weekly weight gain from the second trimester to the third trimester was measured at 0.5 ± 0.2 kilograms per week. This rate remained relatively consistent regardless of whether participants were classified as having a normal, underweight, overweight, or obese status. We assessed the appropriateness of each participant's weight gain during pregnancy by comparing their weekly weight gain rate with

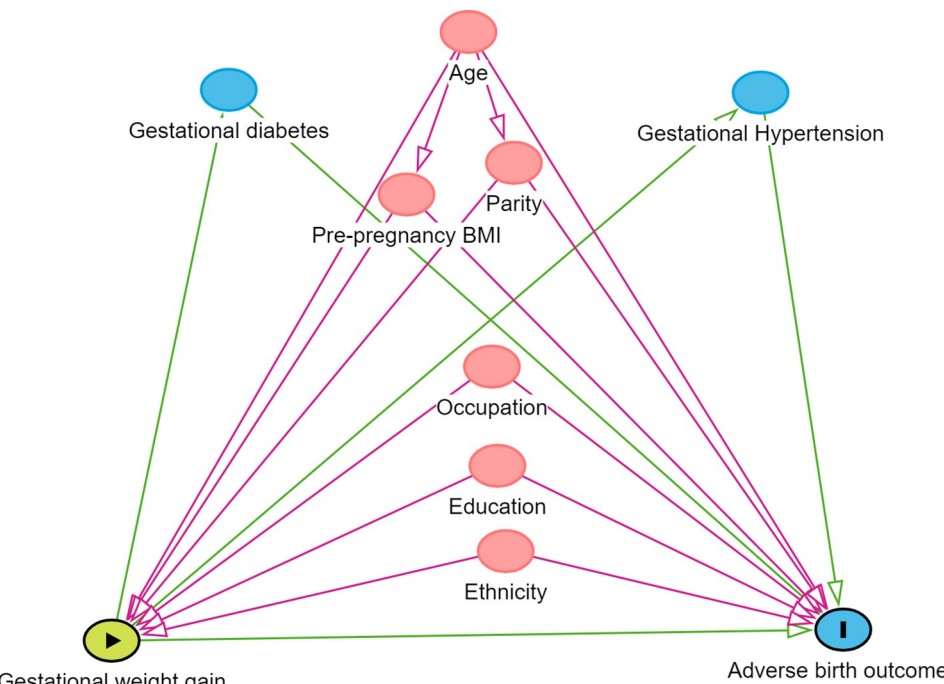

**Fig 1. Directed Acyclic Graph (DAG).** Exposure variable is Gestational weight gain; Outcome variables are adverse birth outcomes (increased infant birth weight and cesarean delivery); Potential confounding variables are age, pre-pregnancy BMI, parity, education, occupation and ethnicity; Potential mediators are gestational diabetes and gestational hypertension.

the recommendations provided by the IOM. These recommendations classify weight gain rates based on the participant's BMI prior to becoming pregnant. The evaluation of GWG adequacy based on pre-pregnancy BMI revealed that slightly over half of the participants (52.4%) experienced a higher-than-recommended GWG rate. In a nearly equal proportion, some participants had either a GWG rate that aligned with the recommendations (26.2%) or a GWG rate that fell below the recommendations (21.5%) (Table 2).

## Birth outcomes

The average birth weight of the infants was recorded as 2964.9 ± 407.0 grams. Approximately 12% of the newborns had a birth weight that was considered low. Additionally, close to 45% of the participants underwent cesarean delivery (Table 3).

## Relationship between gestational weight gain rate and infant birth weight

A significant association was identified between the rate of weight gain during pregnancy and the birth weight of the infants. For each additional 1 kilogram per week of gestational weight gain, the average birth weight of the neonate increased by 392 grams (β = 391.9, 95%CI = (67.2, 716.7), p = 0.02). This relationship was determined while accounting for factors such as the mother's age, education, ethnicity, occupation, parity, and pre-pregnancy BMI. Furthermore, women who experienced a higher-than-recommended rate of GWG (excessive GWG) saw an average increase of 148 grams in the birth weight of their infants compared to women who had a weight gain rate within recommended levels (adequate GWG). This increase was statistically significant (β = 148.1, 95% CI = (8.7, 287.5) grams, p-value of 0.037) (Table 4).

**Table 1. Socio-demographic, obstetric and clinical characteristics of the participants (n = 191).**

| Socio-demographic characteristics | | n(%) |
|---|---|---|
| Age mean±SD, in years | | 26.0 ± 4.2 years |
| Ethnicity | Brahmin/Chettri | 77 (40.3) |
| | Newar | 71 (37.2) |
| | Magar/Tamang/Kami/Damai | 43 (22.5) |
| Religion | Hindu | 169 (88.5) |
| | Buddhist/Christian | 22 (11.5) |
| Education mean±SD | | 12.4 ± 3.4 years |
| Occupation | Homemaker | 95 (49.7) |
| | Business | 39 (20.4) |
| | Service (gov and non-gov) | 24 (12.6) |
| | Others (farmer, teacher, labor, etc) | 33 (17.3) |
| Income | < $290 per month | 37 (19.4) |
| | ≥ $290 per month | 100 (52.4) |
| **Obstetric characteristics** | | |
| Gravida | Primigravida | 82 (42.9) |
| | Multigravida | 109 (57.1) |
| Parity | Nullipara | 108 (56.5) |
| | Multipara | 83 (43.5) |
| Gestational Diabetes | | 7 (3.7) |
| Gestational Hypertension | | 22 (11.5) |
| Hypothyroidism | | 9 (4.7) |
| Gestational weeks at delivery mean±SD | | 38.5 ± 1.1 weeks |
| Pre-pregnancy BMI mean±SD | Total | 24.6 ± 3.7kg/m$^2$ |
| BMI Category | | n(%) |
| | Underweight (18.5kg/m$^2$) | 4 (2.1) |
| | Normal weight (18.5–24.91kg/m$^2$) | 115 (60.2) |
| | Overweight (25.0–29.9kg/ m$^2$) | 55 (28.8) |
| | Obese (≥30kg/m$^2$) | 17 (8.9) |

SD = Standard Deviation, BMI = Body Mass Index, n = frequency.

### Relationship between gestational weight gain and cesarean delivery

Table 5 illustrates the relation between the rate of weight gain during pregnancy and the occurrence of cesarean delivery. However, after accounting for factors such as the mother's age, education, ethnicity, occupation, parity, and pre-pregnancy BMI, no noteworthy association was found between gestational weight gain and the likelihood of cesarean delivery.

## Discussion

This study is one of a kind in Nepal to explore the association between GWG rate and pregnancy outcomes. The mean GWG rate was 0.5 ± 0.2 kg per week. Around fifty percent of the participants experienced GWG that exceeded the recommended limit set by the IOM. Again, overweight and obese women had an excessive GWG rate compared to normal weight or underweight women. Those women who had an excessive GWG during pregnancy had babies with higher birth weights. However, we did not find any significant association between GWG and the likelihood of undergoing a cesarean delivery.

**Table 2. Gestational weight gain rate (n = 191).**

| Gestational weight gain rate mean ±SD | Total | 0.5 ± 0.2kg/week |
|---|---|---|
| | Underweight | 0.5 ± 0.2 |
| | Normal weight | 0.5 ± 0.3 |
| | Overweight | 0.4 ± 0.2 |
| | Obese | 0.5 ± 0.2 |
| **Adequacy of gestational weight gain rate category** | | **n(%)** |
| | Adequate weight gain rate (within recommended range according to BMI) | 50 (26.2) |
| | Inadequate weight gain rate (less than recommended range) | 41 (21.5) |
| | Excessive weight gain rate (more than recommended range) | 100 (52.4) |

SD = Standard Deviation, BMI = Body Mass Index, GWG = Gestational Weight Gain.

Recommended GWG rate range: For underweight = (0.44–0.58) kg/week, normal weight = (0.35–0.50) kg/week, overweight = (0.23–0.33) kg/week, obese = (0.17–0.27) kg/week.

Limited research conducted in low- and middle-income countries has explored the topic of GWG. Among the studies conducted [6,7,13–17], a prevailing trend of excessive GWG among the participants is evident. Studies carried out in Nepal indicated that a significant proportion (49.4%) of participants experienced high rates of GWG [13]. Furthermore, pregnant women categorized as underweight or of normal weight exhibited lower weight gain compared to their overweight or obese counterparts [14]. Correspondingly, a pregnancy cohort in North India revealed that women who were overweight or obese were more prone to surpassing the GWG recommendations set by the IOM, in contrast to underweight or normal-weight women (50.6% and 53.7% vs. 8.8% and 17.9%, respectively) [6]. A comparable pattern emerged from a study conducted in South India [7]. In China, a study reported excessive weight gain rates in 57.9% of pregnant women [15]. In contrast, divergent findings were documented in a Chinese cohort study where women of normal weight and underweight exhibited excessive GWG as opposed to overweight or obese women (78.3% and 12.1% vs. 9.1%, respectively) [16]. In Nigeria, a retrospective study demonstrated mixed results: both underweight (22.2%) and obese (26.2%) women surpassed the recommended IOM levels for GWG [17]. Throughout the body of literature, the majority of studies observed a tendency towards excessive GWG among overweight or obese women, although some studies reported contrasting or mixed outcomes. These variations appear to be influenced by the socio-cultural context of the geographic region under study. Notably, the IOM recommendations were established based on research

**Table 3. Neonatal and maternal outcomes of participants (n = 191).**

| Neonatal and maternal outcomes | | n(%) |
|---|---|---|
| **Birth weight mean±SD** | | 2964.9 ± 407.0 grams |
| | Low birth weight (<2.5kg) | 22 (11.5) |
| | Normal birth weight (2.5–3.9kg) | 167 (87.4) |
| | Macrosomia (≥4.0kg) | 2 (1) |
| **Cesarean delivery** | | 85 (44.5) |

SD = Standard Deviation.

**Table 4. Relationship of gestational weight gain rate with infant birth weight (n = 191).**

| Exposure variables | β (95% CI) | p-value | Adjusted β (95% CI) | p-value |
|---|---|---|---|---|
| Mean Gestational weight gain rate from 2nd to 3rd trimester | 264.3(22.5,506.0) | **0.032** | 391.9(67.2,716.7) | **0.018** |
| Adequate GWG rate | **Ref** | | **Ref** | |
| Inadequate GWG rate | -173.9 (-313.5, -34.2) | **0.015** | -73.3 (-239.3,92.6) | 0.385 |
| Excessive GWG rate | 188.4(74.9,301.8) | **0.001** | 148.1(8.7,287.5) | **0.037** |

BMI = Body Mass Index, GWG = Gestational Weight Gain, CI = Confidence Interval, Ref = Reference category, β = β coefficient, Statistically significant (p<0.05) at 95% CI.

Gestational weight gain rate = weight in 3rd trimester—weight in 2nd trimester / number of weeks in between.

Adjusted for mother's age, education, ethnicity, occupation, parity, and pre-pregnancy BMI.

involving Caucasian urban populations, which might lead to the establishment of differing normative values across geographies and populations, and also importantly across rural and urban settings. It is worth mentioning that our own study primarily involves participants from urban areas. Furthermore, the utilization of GWG rate as an outcome measure is a relatively underexplored area, with most studies employing total GWG instead. Excessive gestational weight gain poses a significant concern in diverse socio-cultural and geographical contexts, necessitating careful study to mitigate associated risks and adverse outcomes for both mothers and newborns. Healthcare providers must prioritize strategies aimed at preventing excessive GWG to safeguard maternal and neonatal health. This proactive approach can potentially improve pregnancy outcomes and overall well-being during childbirth.

This study revealed that the average birth weight of infants increased by approximately 392 grams for every 1 kg per week rise in maternal gestational weight between the second and third trimesters. One of the Indian studies demonstrated a significant connection between weight gain in the third trimester and newborn birth weight (P = 0.022), irrespective of the mother's BMI [9]. Similarly, a study conducted in Nepal established a positive correlation between pregnancy weight gain in women and newborn birth weight, indicating that maternal GWG played a pivotal role in predicting infant birth weight [14]. In comparison to women with a normal rate of GWG, neonates born to those with excessive rates had an average birth weight increase of 148 grams in this study. This finding was supported by a Chinese study that reported an adjusted odds ratio of 5.39 (95% CI 2.94 to 9.89; p = 0.001) for high birth weight infants among mothers with excessive GWG compared to those with adequate weight gain [18]. An Indian systematic review demonstrated that women with excessive GWG faced an elevated risk of delivering high birth weight babies [5]. In a Chinese birth cohort, excessive GWG during later pregnancy stages was associated with a 150% higher likelihood of giving

**Table 5. Relationship of gestational weight gain rate with cesarean delivery (n = 191).**

| Exposure variables | Univariate OR (95% CI) | p-value | Multivariate AOR (95% CI) | p-value |
|---|---|---|---|---|
| Mean Gestational weight gain rate from 2nd to 3rd trimester | 1.7(0.5–5.9) | 0.356 | 1.3(0.3–4.7) | 0.687 |
| Adequate GWG | **Ref** | | **Ref** | |
| Inadequate | 0.4(0.1–0.9) | **0.026** | 0.5(0.1–1.1) | 0.093 |
| Excessive GWG | 0.9(0.4–1.8) | 0.908 | 0.8(0.4–1.7) | 0.603 |

BMI = Body Mass Index, GWG = Gestational Weight Gain, CI = Confidence Interval, Ref = Reference category, OR = Odds Ratio, AOR = Adjusted Odds Ratio, *Statistically significant (p<0.05) at 95% CI, Gestational weight gain rate = weight in 3rd trimester—weight in 2nd trimester / number of weeks in between Adjusted for mother's age, education, ethnicity, occupation, parity, and pre-pregnancy BMI.

birth to a large-for-gestational-age (LGA) baby [19]. Similarly, a study in Brazil indicated a positive relationship between excessive maternal weight gain and both percentage of body fat and birth weight in offspring [20]. The underlying mechanism can be attributed to a positive energy balance experienced by pregnant women with excessive weight gain which results in energy accumulation, heightened insulin resistance, and increased transport of glucose and fatty acids across the placenta. These factors collectively contribute to amplified fetal growth, augmented deposition of fetal fat, and synthesis of leptin within adipocytes [20,21]. Notably, high birth weight or macrosomia in infants is associated with various maternal and neonatal complications [22–24]. Consequently, these findings underscore the significance of offering pregnant women guidance that fosters a comprehensive understanding of weight gain, encourages sensible weight management, and diminishes the potential for adverse neonatal outcomes linked to excessive birth weight.

This study did not find any relation between GWG and CS. Similar findings were reported in studies done in Iran and America [25,26]. But various studies done in Asian countries show a significant association between GWG and CS. In India, a study showed that obese women who exceeded the recommended weight gain had a greater risk of undergoing a CS (odds ratio: 1.9, 95% confidence interval: 1.4–2.5; p < 0.001) [7]. A study in China iterated that participants in the CS group generally had a higher rate of excessive GWG (33.0% vs. 23.4%, P 0.001) compared to the vaginal delivery group [8]. A significant association between GWG rate and CS was also seen in Korean women [27]. A large population-based cohort study in China also found that excessive GWG was associated with higher risk of CS [28]. Offering preconception and prenatal counseling to achieve and sustain the recommended weight gain becomes essential for ensuring optimal maternal-infant health outcomes [29]. With the rising rate of cesarean delivery in Nepal [11,12], screening populations with a higher likelihood of CS is crucial. Similar study with larger sample size may help to identify the actual relation between GWG and cesarean delivery in Nepal. Also, the alarming rate of rising cesarean delivery demands for a thorough study into the indications for these procedures and whether they are genuinely warranted.

## Strengths and limitations

This study possesses notable strengths that merit attention. It stands out as one of the pioneering investigations in Nepal to explore the link between GWG rate and both neonatal birth weight and the likelihood of cesarean delivery. The measurement of pregnant women's weights during all trimester antenatal visits was conducted by trained research nurses, while neonatal anthropometric data were extracted from hospital medical records, ensuring the reliability of the dataset. Moreover, overweight or obese participants received dietary and physical activity guidance in line with established hospital protocols, aimed at mitigating unfavorable pregnancy outcomes.

Nevertheless, we acknowledge the limitations inherent in this study. Firstly, for assessing pre-pregnancy BMI, reliance was placed on weight measurements collected during the initial prenatal visit. This approach was adopted due to the impracticality of self-reported pre-pregnancy weight, given the absence of home or nearby health facility weight measurement resources. Despite the inherent limitations of this method, there were no viable alternatives unless pre-pregnancy weight data could be gathered prospectively before pregnancy. Consequently, early pregnancy weight measurements were utilized. Secondly, since the study was confined to an urban setting, there exists the possibility that the sample might not accurately represent the rural population or other socio-economic strata. Thirdly, the categorization of GWG based on BMI was executed following the guidelines laid out by the IOM [4]. However,

recent research has begun to question the applicability of the IOM guidelines for GWG. A primary concern is that these guidelines might not be universally relevant to other racial and ethnic groups, primarily because they were developed using the Caucasian standard [7,30–32]. Lastly, a key limitation of this study is that it reports findings from a single center in Nepal, which may not represent the broader demographic, cultural and medical variations found in other regions or country.

## Conclusion

This study conducted among pregnant women in Nepal demonstrated a significant association between the average rate of GWG from the second to the third trimester and the birth weight of neonates. The strength of this association varied based on the pre-pregnancy BMI of the mothers. Notably, overweight or obese women exhibited a higher rate of GWG, and excessive weight gain emerged as a key predictor of neonatal birth weight. No significant association was found between the GWG rate and cesarean delivery. Subsequent research with larger sample sizes and an extended follow-up period is essential to assess how GWG affects the birth weight of newborns, particularly in relation to obesity and cardiovascular diseases and its impact on cesarean delivery. Ongoing research endeavors should persist in exploring other modifiable elements contributing to suboptimal GWG among pregnant women in Nepal. These investigations will be valuable in shaping culturally adapted dietary and lifestyle interventions and guidelines specifically designed for this population. Considering the crucial necessity to tackle and break the pattern of escalating weight gain and negative health outcomes, it is imperative that early intervention before pregnancy becomes a primary focus of future public health endeavors.

## Acknowledgments

We thank all research staffs, nursing staffs and medical personnel who helped with data collection. We are grateful to the participants and their families.

## Author Contributions

**Conceptualization:** Jyoti Nepal, Shristi Rawal.

**Formal analysis:** Jyoti Nepal.

**Investigation:** Jyoti Nepal, Kalpana Chaudhary, Shristi Rawal.

**Methodology:** Jyoti Nepal, Archana Shrestha, Shristi Rawal.

**Project administration:** Jyoti Nepal, Kalpana Chaudhary.

**Resources:** Jyoti Nepal.

**Software:** Bikram Adhikari.

**Supervision:** Jyoti Nepal, Abha Shrestha, Archana Shrestha, Sangita Pudasainee Kapri, Shristi Rawal.

**Validation:** Jyoti Nepal, Bikram Adhikari, Archana Shrestha, Sangita Pudasainee Kapri, Shristi Rawal.

**Visualization:** Jyoti Nepal, Shristi Rawal.

**Writing – original draft:** Jyoti Nepal.

**Writing – review & editing:** Kalpana Chaudhary, Bikram Adhikari, Abha Shrestha, Archana Shrestha, Sangita Pudasainee Kapri, Shristi Rawal.

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
