## [Decision Letter · Decision Letter 0]

28 May 2024

PGPH-D-24-00323

Association of gestational weight gain rate with infant birth weight and cesarean delivery: A prospective cohort study in Nepal

Dear Dr. Nepal,

Thank you for submitting your manuscript to PLOS Global Public Health. After careful consideration, we feel that it has merit but does not fully meet PLOS Global Public Health’s publication criteria as it currently stands. Therefore, we invite you to submit a revised version of the manuscript that addresses the points raised during the review process.

The study reports findings from a single center in Nepal. The study has included only women who underwent LSCS and hence the results may not be generalizable to all the antenatal women. Please add this one of the limitations and address the reviewer's comments.

We look forward to receiving your revised manuscript.

Kind regards,

Leeberk Raja Inbaraj, MD

Academic Editor

Journal Requirements:

Additional Editor Comments (if provided):

Reviewers' comments:

Reviewer's Responses to Questions

**Comments to the Author**

1. Does this manuscript meet PLOS Global Public Health’s publication criteria? Is the manuscript technically sound, and do the data support the conclusions? The manuscript must describe methodologically and ethically rigorous research with conclusions that are appropriately drawn based on the data presented.

Reviewer #1: Yes

2. Has the statistical analysis been performed appropriately and rigorously?

Reviewer #1: Yes

3. Have the authors made all data underlying the findings in their manuscript fully available (please refer to the Data Availability Statement at the start of the manuscript PDF file)?

Reviewer #1: Yes

4. Is the manuscript presented in an intelligible fashion and written in standard English?

Reviewer #1: Yes

5. Review Comments to the Author

Reviewer #1: Association of gestational weight gain rate with infant birth weight and cesarean delivery: A prospective cohort study in Nepal

PGPH-D-24-00323

Overall comment:

This is an important study in Nepal that has studied the relationship between gestational weight gain and birth weight, and further cesarean section. The paper adds value to the Nepalese population given that the rate of cesarean deliveries is alarmingly high in the population.

Abstract:

This is well written. It is important to comment on association of GWG with c-section in the conclusion as well, and since there was no association, well powered studies are needed further to study the same.

Introduction:

Line 63: GWG can be used instead as it has already been expanded (similarly in line 65, 66, 67 and 79)

Line 64: Kindly replace “in connection” with “in relation”

Line 71: Can be modified as “..whereas those with inadequate GWG had a higher risk of delivering low birth weight, small for gestational age, and preterm born infants”

Line 71: Instead of ‘neonatal birth weight’, it can be referred to as plain ‘Birth weight’

Line 74: It would be useful to have the current cesarean rate in Nepal and state if it is >10%, given that cesarean section rates >10% are not associated with reductions in maternal and newborn mortality rates: https://www.who.int/news-room/questions-and-answers/item/who-statement-on-cesarean-section-rates-frequently-asked-questions

Line 78: Kindly replace “become crucial” with “is crucial.

Line 79: Remove “the”

Materials and methods:

Line 90: Modify to “recruitment of participants” instead of “participants recruitment”

Line 97: OPD is not required elsewhere in the manuscript, so this can be omitted

Line 99: Modify to: “Overall, 191 participants who had all the required antenatal visits with liveborn, term delivery were included in the analysis.”

Line 111: Modify to: “Face-to-face interviews were conducted..”

Line 113:I think it should be referred to as ‘anthropometric’ and not ‘clinical’ variables

Line 117: GWG alone can be used and expansion is not needed here

Figure 1: Please add a footnote to understand if the pink and green lines in the DAG represent anything specific

Results:

Line 147: BMI has been expanded in Introduction, so sufficient to use BMI throughout further on”

Line 148: Modify to: “The majority of participants (60.2%) had normal BMI status prior to pregnancy”

Table 1:

• Was a standard classification used for occupation and income (?median) – if yes, please reference/mention the same in methods

• Were there women who had both gestational Diabetes and Hypertension

Table 3:

• Remove the extra bracket

• Were indications for the 85 cesarean deliveries studied? It would be good to indicate those and correlate the same with excessive GWG.

Discussion:

Line 212: Modify to “Those women who had an excessive GWG…”

Line 214: Replace “connection” with “significant association”

Use GWG across uniformly

Line 219: Replace “elevated” with “high”

Line 236: To rephrase – “..which might lead to the establishment of differing normative values across geographies and populations, and also importantly across rural and urban settings.”

Line 243-241: This sentence brings in some disconnect, and if a recommendation, needs to brought in later.

Line 270: It is important that a careful recommendation be made – it is possible that along with GWG would come along other associated and concurrent risk factors such as Gestational diabetes, hypertension, lifestyle factors, etc. and emphasis be laid on a more holistic approach an screening high risk cases and counselling (?pre-conceptional counselling). Also, it is perhaps important to reiterate a careful study on the increasing rate of c-sections in Nepal to identify other causes for which c-sections are being indicated, and if the indications truly warrant a c-section.

6. PLOS authors have the option to publish the peer review history of their article (what does this mean?). If published, this will include your full peer review and any attached files.

**Do you want your identity to be public for this peer review?** For information about this choice, including consent withdrawal, please see our Privacy Policy.

Reviewer #1: No

---

## [Editor Report · Decision Letter 1]

8 Oct 2024

Association of gestational weight gain rate with infant birth weight and cesarean delivery: A prospective cohort study in Nepal

PGPH-D-24-00323R1

Dear Nepal,

We are pleased to inform you that your manuscript 'Association of gestational weight gain rate with infant birth weight and cesarean delivery: A prospective cohort study in Nepal' has been provisionally accepted for publication in PLOS Global Public Health.

Best regards,

Leeberk Raja Inbaraj, MD

Academic Editor